# Cobalt Protoporphyrin Downregulates Hyperglycemia-Induced Inflammation and Enhances Mitochondrial Respiration in Retinal Pigment Epithelial Cells

**DOI:** 10.3390/antiox14010092

**Published:** 2025-01-15

**Authors:** Peng-Hsiang Fang, Tzu-Yu Lin, Chiu-Chen Huang, Yung-Chang Lin, Cheng-Hung Lai, Bill Cheng

**Affiliations:** 1Department of Veterinary Medicine, National Chung-Hsing University, Taichung 402, Taiwan; 2Graduate Institute of Biomedical Engineering, National Chung-Hsing University, Taichung 402, Taiwan; 3Department of Post-Baccalaureate Veterinary Medicine, Asia University, Taichung 413, Taiwan; 4Department of Animal Health, Hungkuang University, Taichung 433, Taiwan

**Keywords:** cobalt protoporphyrin, diabetic retinopathy, hyperglycemia, ARPE-19, mitochondria

## Abstract

Diabetic retinopathy is characterized by hyperglycemic retinal pigment epithelial cells that secrete excessive pro-inflammatory cytokines and VEGF, leading to retinal damage and vision loss. Cobalt protoporphyrin (CoPP) is a compound that can reduce inflammatory responses by inducing high levels of HO-1. In the present study, the therapeutic effects of CoPP were examined in ARPE-19 cells under hyperglycemia. ARPE-19 cells were incubated in culture media containing either 5.5 mM (NG) or 25 mM (HG) glucose, with or without the addition of 0.1 µM CoPP. Protein expressions in samples were determined by either Western blotting or immunostaining. A Seahorse metabolic analyzer was used to assess the impact of CoPP treatment on mitochondrial respiration in ARPE-19 cells in NG or HG media. ARPE-19 cells cultured in NG media displayed different cell morphology than those cultured in HG media. CoPP treatment induced high HO-1 expressions and significantly enhanced the viability of ARPE-19 cells under hyperglycemia. Moreover, CoPP significantly downregulated expressions of inflammatory and apoptotic markers and significantly upregulated mitochondrial respiration in APRPE-19 cells under hyperglycemia. CoPP treatment significantly enhanced cell viability in ARPE-19 cells under hyperglycemia. The treatment also downregulated the expressions of pro-inflammatory and upregulated mitochondrial respiration in the hyperglycemic cells.

## 1. Introduction

Diabetic retinopathy (DR) is a diabetes-related eye disease that affects the retina. It is widely recognized as the leading cause of diabetes-related vision impairment or loss among working-age adults and older adults worldwide [1]. It has been estimated that by 2030, the number of patients affected by DR will increase to 191 million. A key characteristic of DR is the increased production of reactive oxygen species (ROS) in the retinal pigment epithelium (RPE). This oxidative stress causes damage to cellular components like lipids, proteins, and DNA. Additionally, mitochondrial respiration and biogenesis are downregulated in DR patients’ RPE cells [2]. The damaged RPE cells secrete inflammatory cytokines, contributing to local inflammation and retinal damage. Moreover, damaged RPE cells can increase the production of VEGF, which contributes to the abnormal growth of fragile blood vessels, causing further retinal damage and vision loss [3].

Laser therapy and anti-VEGF injections are the two common clinical treatments for DR [4]. Laser therapy, such as panretinal photocoagulation, is used for proliferative DR, where abnormal blood vessels grow in the retina. The laser shrinks these vessels and reduces the risk of retinal detachment and vision loss [5]. However, in addition to the need for expensive equipment, panretinal photocoagulation could induce permanent loss of peripheral vision, as the treatment reduces blood flow to areas of the retina where the laser is applied [5]. Moreover, panretinal photocoagulation can sometimes lead to or worsen swelling in the central retina, also known as macular edema, which can cause blurring of the central vision [6]. Similarly, the cost of anti-VEGF injections can become a financial burden to the patients. Anti-VEGF injections must be administered regularly every 4–8 weeks [7]. The need for ongoing injections over months or years can be burdensome for patients and may lead to treatment fatigue. Hence, patients may struggle with adherence to the recommended injection schedule due to the frequency of visits, discomfort, or logistical challenges, which can reduce the effectiveness of the treatment [7].

Anti-inflammatory agents have also been demonstrated to be a good alternative for treating DR through the downregulation of inflammatory cytokine production in RPE cells and mitigation of the breakdown of the RPE layer [8]. Moreover, anti-inflammatory treatment can complement anti-VEGF therapy, as hyperglycemia-induced inflammations in RPE cells can cause the cells to secrete VEGF, which triggers neovascularization [8]. Although many anti-inflammatory drugs have been developed, their responses in DR patients vary. Therefore, there has been a growing interest in developing various anti-inflammatory agents for treating DR.

Cobalt protoporphyrin (CoPP) is a potent inducer of heme oxygenase-1 (HO-1), which offers multiple therapeutic benefits through its enzymatic activity. HO-1 breaks down heme into biliverdin, carbon monoxide (CO), and iron, which is crucial for regulating cellular oxidative stress and iron homeostasis and generating signaling molecules such as carbon monoxide, which has vasodilatory and anti-inflammatory properties [9]. Several in vivo studies have proven that CoPP has cytoprotective effects in ischemic liver tissues [10], myocardial infarction [11], and skin wound healing [12], but not in hyperglycemia-damaged RPE. Accordingly, the therapeutic effect of CoPP in RPE cells under hyperglycemia was investigated in the present study. ARPE-19, a human RPE cell line [13], was used to examine the cytoprotective effects of CoPP under either normoglycemia or hyperglycemia.

## 2. Materials and Methods

### 2.1. Cell Culture

Human retinal pigment epithelial cells, ARPE-19s (Elabscience, Houston, TX, USA, Cat. # EP-CL-0026), were cultured in DMEM/F-12 (Gibco, Taipei, Taiwan, Cat. # 12500062) supplemented with 5.5 mM or 25 mM of glucose, 10% fetal bovine serum (Gibco, Cat. # 16000044), and 1% penicillin-streptomycin (Gibco, Cat. # 15140122) at 37 °C in a 5% CO_2_ humidified incubator. The culture medium was changed every 2–3 days. Approximately 0.1 µM of cobalt protoporphyrin (CoPP, Enzo Life Science, Taipei, Taiwan, Cat. # ALX-430-076-M025) was added to cell cultures designated as CoPP-treated samples for at least 12 h. The working concentration of 0.1 µM was identified as the best concentration to work with, according to a previous study on the therapeutic effect of CoPP in keratinocytes [12]. The numbers of viable cells were determined by Trypan blue staining and a hemocytometer. All cell images were taken with an inverted fluorescence microscope (Nexcope, Ningbo, China, Cat. # NIB620-FL).

### 2.2. Immunofluorescence Staining

The cells were first seeded onto glass slides and incubated at 37 °C overnight. The next day, the cells were exposed to 0.1 µM CoPP in either DMEM-low or DMEM-high glucose at 37 °C. After PBS washing, the cells were fixed with dehydrated methanol at 4 °C for 10 min. The fixed cells were then washed with cold PBS and incubated with anti-ZO-1 antibodies (Sigma-Aldrich, Taipei, Taiwan, Cat. # ZRB1761) at 4 °C, shaking overnight. Approximately 0.1% PBST was used to wash away any unbound anti-ZO-1 antibodies, and the cells were then incubated with FITC-conjugated secondary antibodies in the dark at room temperature for one hour. The cells were washed with cold 0.1% PBST and imaged with a fluorescence microscope.

### 2.3. Dodecyl Sulfate Polyacrylamide Electrophoresis (SDS-PAGE) and Western Blotting

SDS (4–12%) polyacrylamide gels (ThermoFisher Scientific, Taipei, Taiwan, Cat. # HC2040) were prepared according to the manufacturer’s protocol using Mini-PROTEAN^®^ Tetra Handcast Systems (Bio-Rad, Taipei, Taiwan, Cat# 1658000FC). Samples were electrophorized at 100 V at room temperature until the dye front had reached the bottom of the gel. Subsequently, the samples were transferred onto a 0.45 µm polyvinylidene fluoride membrane (Merk Millipore, Taipei, Taiwan, Cat# IPVH00010). After blocking overnight at 4 °C, the membrane was exposed to primary antibodies at 4 °C, shaking for another overnight. It was washed with 0.1% Tris-buffered saline with 0.1% Tween^®^ 20 detergent (TBST) before applying secondary antibodies for 1 h at room temperature with shaking. After washing with 0.1% TBST, the signals were detected with ECL-plus reagent (Nacalai Tesque, Kyoto, Japan, Cat. # 07880-70) and imaged using the Odyssey Fc imaging system (LI-COR Biosciences). The list of primary and secondary antibodies is listed in Appendix A.

### 2.4. Mitochondrial Respiration

The mitochondrial respirations of each sample were measured with a Seahorse metabolic analyzer (Agilent, Santa Clara, CA, USA), and the experiment’s design was based on previously published protocols [14]. Cells were seeded in a Seahorse cell culture plate (Agilent, Cat. # 100777-004) with DMEM F-12 media containing either 5.5 mM or 25 mM glucose for one overnight. The next day, the cells were treated with 0.1 µM CoPP at 37 °C for 12 h before loading the plate into the analyzer. Each well contains four drug injection ports located on the probe disc, which were loaded with 1.5 μM oligomycin (Merck, Taipei, Taiwan, Cat. # 495455), 0.5 μM FCCP (Merck, Cat. # C2920), 0.5 μM antimycin A (Merck, Cat.# 1397-94-0), and 0.5 μM rotenone (Merck, Cat.# 557368). The four drugs were injected into all wells per the manufacturer’s protocol.

Oligomycin inhibits ATP synthase (complex V), which reduces mitochondrial respiration linked to cellular ATP production. FCCP is an uncoupling agent that collapses the proton gradient and disrupts the mitochondrial membrane potential. It was injected after oligomycin. As a result, electron flow through the ETC was uninhibited, and oxygen consumption by complex IV reached the maximum. The FCCP-stimulated oxygen consumption rate could then be used to calculate spare respiratory capacity. The last injection was a mixture of rotenone, a complex I inhibitor, and antimycin A, a complex III inhibitor. This combination shuts down mitochondrial respiration and enables the calculation of nonmitochondrial respiration driven by processes outside the mitochondria.

### 2.5. Statistical Analysis

A one-way analysis of variance test and then a Tukey post hoc test were used to compare the statistical significance between treatments under test conditions. A *p* value of less than 0.05 was considered significant.

## 3. Results

### 3.1. Hyperglycaemia-Induced Morphological Changes in ARPE-19 Cells

ARPE-19 cells were cultured in conditioned media that contained either 5.5 mM (normoglycemia, NG) or 25 mM (hyperglycemia, HG) glucose, and changes in cell morphology over 8 days were monitored (Figure 1). Distinct differences in cell morphology were noticed from day 5 onwards and became more evident on day 8. In NG media, ARPE-19 cells displayed elliptical shapes (red arrows), whereas spindle shapes were noticed in the cells cultured in HG media (blue arrows). Surprisingly, CoPP induced cellular stress in normoglycemic ARPE-19 cells, evidenced by the formation of cell debris atop a confluent layer of ARPE-19 cells (yellow arrows). This phenomenon was not observed in hyperglycemic ARPE-19 cells treated with CoPP. Instead, CoPP treatment promoted higher cell density (green arrows), suggesting that under hyperglycemic conditions, CoPP could promote ARPE-19 cell proliferation. However, it was noticed on day 8 that the CoPP-treated ARPE-19 cells under hyperglycemia lost their cobblestone morphology and displayed fibroblast-like structure. In comparison, the CoPP-treated ARP-19 cells under normoglycemia retained their cobblestone morphology but developed multiple tunneling nanotubes after 8 days of culturing.

### 3.2. Viable Cell Count and Viability

The positive effect of CoPP on cell proliferation was further confirmed by analyzing the number of viable cells (Figure 2A). There was no difference in viable cell numbers between the NG and HG groups throughout the 8 days of culturing. However, the exposure to CoPP as early as day 2 resulted in a significant increase in viable cell numbers, regardless of whether in NG or HG media. Interestingly, the number of ARPE-19 cells that received CoPP treatment while in HG media was significantly higher than that cultured in NG media with CoPP treatment. This indicated that glucose concentration could influence the function of CoPP.

Interestingly, when considering the number of dead cells, the viability of ARPE-19 cells cultured in HG media was significantly lower than those cultured in NG media (Figure 2B). Moreover, the CoPP treatment significantly reduced cell viability in the normoglycemic cell culture. The viability of hyperglycemic ARPE-19 cells that received CoPP treatment was markedly higher than those not exposed to CoPP. Therefore, CoPP could promote ARPE-19 cell proliferation and enhance their viability under hyperglycemia. Under normoglycemia, CoPP also promoted cell proliferation but also had detrimental effects on ARPE-19 cell viability.

### 3.3. Western Blotting Analysis

Western blotting analysis also confirmed the positive effects of CoPP on ARPE-19 cell proliferation (Figure 3A). As expected, CoPP induced significantly higher HO-1 and Nrf2 protein expressions in ARPE-19 cells cultured in either NG or HG media (Figure 3B,C), suggesting the Nrf2/HO-1 pathway had been activated in both culturing conditions. In the absence of CoPP treatment, expressions of cell proliferation proteins Ki-67 and TGFβ1 were significantly lower than those treated with CoPP (Figure 3D,E). Expressions of inflammatory, angiogenic, and apoptotic protein markers under the influences of hyperglycemia were also analyzed (Figure 3F). ARPE-19 cells cultured in HG media had significantly higher IL-1β and TNFα expressions than those cultured in NG media. In contrast, CoPP treatment significantly reduced the expressions of the two inflammatory markers in hyperglycemic ARPE-19 cells (Figure 3G,H). Similarly, the expressions of VEGF were significantly higher in the cells cultured in HG media, and CoPP treatment significantly reduced the angiogenic protein expressions (Figure 3I). Surprisingly, CoPP treatment significantly attenuated the expressions of activated caspase-3 in ARPE-19 cells under hyperglycemia but displayed high activated caspase-3 expressions in the cells under normoglycemia (Figure 3J). Thus, CoPP treatment significantly reduced hyperglycemia-induced inflammatory, angiogenic, and apoptotic protein expressions. However, under normoglycemia, CoPP treatment induced high expression of activated caspase-3 in ARPE-19 cells. This suggested that glucose concentration could modulate the apoptotic activity of CoPP in ARPE-19 cells.

### 3.4. Immunostaining

The hyperglycemia-induced damages in tight junctions were evaluated via the immunostaining of junction proteins, zonula occludens-1 (ZO-1). ARPE-19 cells were exposed to either CoPP-containing NG or HG media overnight, and their nucleus and ZO-1 proteins were immunostained the next day (Figure 4A). It was expected that no difference would be seen in DAPI staining of the nucleus among all four treatment groups. However, the DAPI signals in the HG group were significantly lower than in the NG group (Figure 4B). Likewise, the normoglycemic cells treated with CoPP displayed weak DAPI signals. In contrast, under hyperglycemia, CoPP significantly enhanced the DAPI signals in ARPE-19 cells, as opposed to those not treated with CoPP. Similar results were also noticed in ZO-1 staining, where normoglycemic ARPE-19 cells had significantly higher ZO-1 signals than the hyperglycemic cells. (Figure 4C). Compared to the cells cultured in NG media without CoPP, adding CoPP did not affect ZO-1 expression under normoglycemia. In contrast, CoPP significantly enhanced the ZO-1 expressions in ARPE-19 cells cultured in HG media. Accordingly, CoPP treatment could attenuate the hyperglycemia-induced damage in ARPE-19 cells.

### 3.5. Analysis of Mitochondrial Respiration

Since hyperglycemic-induced damage is closely associated with mitochondrial function, the effect of CoPP treatment on mitochondrial respiration under either normoglycemia or hyperglycemia was assessed with a Seahorse metabolic analyzer (Figure 5A). The 75 min measurement showed that ARPE-19 cells cultured in CoPP-containing NG or HG media displayed a higher oxygen consumption rate (OCR) than their no CoPP treatment counterpart. Analyzing the last time point of the basal respiration phase, no significant difference was noticed in the ARPE-19 cells cultured in either NG or HG media. However, CoPP treatment significantly increased the OCR measurement in ARPE-19 cells cultured in either medium (Figure 5B). A significant difference was noticed between the NG and HG groups in the ATP-linked production phase, as the hyperglycemic cells displayed significantly less OCR (Figure 5C). Like the basal respiration phase, CoPP treatment significantly improved the mitochondrial respiration in ARPE-19 under both glucose conditions in the ATP-linked production phase. In the maximum respiration phase, a significant difference in the OCR was noticed between the cells of the NG and HG groups. CoPP treatment only significantly enhanced the respiratory activity in ARPE-19 cells cultured in the HG group (Figure 5D). In the spare respiratory activity phase, ARPE-19 cells had substantially higher OCR readings after receiving CoPP treatment in both glucose media (Figure 5E). These data indicated that CoPP could rejuvenate mitochondrial respiration in hyperglycemic APRE-19 cells.

### 3.6. CoPP Promotes Mitochondrial Biogenesis

It was postulated that the anti-inflammatory effects exerted by CoPP in hyperglycemic ARPE-19 cells, which resulted in increased mitochondrial respiration, could be due to increased mitochondrial biogenesis. To demonstrate that CoPP could induce mitochondrial biogenesis, cell lysates of ARPE-19 cells cultured in NG or HG media, with or without CoPP, were prepared. Subsequently, the expression levels of protein markers that promote mitochondrial biogenesis were examined (Figure 6A). There were low expression levels of PGC-1α in ARPE-19 cells cultured in either NG or HG media. The addition of CoPP resulted in a significant increase in the expression of the protein marker under either glucose condition (Figure 6B). The expression of Nrf1 was significantly higher in ARPE-19 cells under hyperglycemia than in those cultured under normoglycemia, and CoPP significantly elevated the expression under both glucose conditions (Figure 6C). Interestingly, the expression levels of mtTFA in the cells cultured in NG media were significantly higher than those cultured in HG media or NG media with CoPP (Figure 6D). The addition of CoPP, however, significantly increased mtTFA expression level under hyperglycemia. Collectively, CoPP was demonstrated to enhance mitochondrial biogenesis in ARPE-19 cells under hyperglycemia.

## 4. Discussion

The most surprising finding in the present study was the morphological changes seen in ARPE-19 cells after exposure to different treatments. Structurally, healthy individual RPE cells within an RPE layer displayed hexagonal shapes [15], whereas the same shapes are still retained under hyperglycemia, except tight junctions are disrupted [16]. In the present study, however, ARPE-19 cells cultured in NG media displayed elliptical shapes with cobblestone morphology. In contrast, spindle shapes with cobblestone morphology were noticed when the cells were cultured in HG media, which was also seen in a previous study [17]. The physiological significance or relevance of such morphological difference is not understood, but it was progressive at least up to 8 days of culturing. Hence, the data indicated that ARPE-19 cells do not share the same cell morphology as primary RPE cells under either normoglycemia or hyperglycemia. Likewise, the morphological changes in CoPP-treated ARPE-19 cells under either normoglycemia or hyperglycemia were unexpected. Although CoPP promoted cell proliferation under hyperglycemia, the cobblestone morphology disappeared after 8 days of treatment. This suggested long-term exposure to CoPP was detrimental to RPE cells since cobblestone morphology is a positive sign of healthy RPE cells [18]. Long-term exposure to CoPP probably led to the overexpression of HO-1, which has been shown to promote pulmonary fibrosis [19]. Since the cobblestone morphology was still visibly seen in hyperglycemic ARPE-19 cells on day 5 with CoPP treatment, continuous CoPP exposure for 5 days was sufficient to promote ARPE-19 cell proliferation without compromising their optimal cell morphology.

Another interesting finding was the effect of CoPP on normoglycemic ARPE-19 cells. Unlike the CoPP-treated ARPE-19 cells in HG media, cobblestone morphology was noticeable in the CoPP-treated ARPE-19 cells under normoglycemia but with significantly low cell viability. Moreover, the visibility of tunneling nanotubes forming between the cells suggested that the CoPP treatment was causing cell stress in normoglycemic ARPE-19 cells [20]. Despite the detrimental effects on the normoglycemic cells, the CoPP treatment still significantly increased ARPE-19 cell proliferation under normoglycemia for up to 5 days, but there was no significance on day 8 compared to those not CoPP-treated. Accordingly, the bioactivities of CoPP in ARPE-19 cells were clearly under the influence of glucose concentration in the media. This glucose-dependent bidirectional effect of CoPP has also been reported in human keratinocytes, where CoPP treatment enhanced the cell migration under hyperglycemia but not under normoglycemia [12]. Hence, it is believed this glucose-dependent effect probably served as feedback control, slowing the CoPP-induced RPE cell proliferation when the blood glucose level is restored to normal, preventing the chance of developing RPE hyperplasia [21].

Interestingly, the same glucose-dependent changes in cell morphology were not noticed in the protein expressions in ARPE-19 cells, with or without CoPP treatment. The strong expressions of HO-1 and Nrf2 in CoPP-treated ARPE-19 cells indicated that glucose concentration did not affect the CoPP-induced HO-1/Nrf2 pathway activation. The strong expressions of proliferation markers Ki67 and TGFβ1 further supported that CoPP treatment could promote ARPE-19 cell proliferation under both normoglycemia and hyperglycemia. HO-1 is known to promote cell proliferation in melanoma through the B-Raf-ERK signaling pathway [20], thus suggesting that the CoPP-induced HO-1 could have also promoted ARPE-19 cell proliferation through the activation of the same signaling pathway, which led to the overexpression of Ki-67 and TGFβ1. The CoPP-induced HO-1 also led to downregulations of IL-1β and TNFα, two inflammatory markers known to be expressed in RPE cells under hyperglycemia [22]. Studies on gastric cancers claimed HO-1 could promote VEGF expressions [23]. However, the VEGF expressions in hyperglycemic ARPE-19 cells were significantly downregulated when exposed to CoPP treatment. Similar results were also noticed in a clinical study on patients with type 2 diabetes (T2D). It was discovered that T2D patients with proliferative diabetic retinopathy had significantly higher VEGF expressions and significantly lower HO-1 expressions when compared to healthy individuals [24]. Collectively, unlike the changes in cell morphology described above, CoPP-induced HO-1 expressions in ARPE-19 cells were not affected by glucose concentration. Through the HO-1 expressions, the hyperglycemia-induced inflammations in ARPE-19 cells were significantly reduced. Likewise, VEGF expressions in hyperglycemic ARPE-19 cells were downregulated in the presence of CoPP, although the underlying mechanism requires further investigation.

Hyperglycemia-induced activated caspase-3 expressions in ARPE-19 cells were significantly reduced in the presence of CoPP, confirming that CoPP could attenuate hyperglycemia-induced cell apoptosis in ARPE-19 cells. The significant enhancement in DAPI signaling in CoPP-treated ARPE-19 cells in HG media also confirmed the anti-apoptotic effect of the treatment. The nucleus of hyperglycemic cells has low DAPI signals due to hyperglycemic-induced damage in the DNA [25]. The upregulation of activated caspase-3 and low DAPI signals in normoglycemic ARPE-19 cells with CoPP treatment further demonstrated that the treatment induces cell death under normoglycemia but not hyperglycemia. Interestingly, CoPP did not affect the expression pattern of ZO-1 in ARPE-19 cells under normoglycemia, suggesting the glucose-dependent effect of CoPP activity was limited to apoptosis and not tight junction formation. The enhancement of ZO-1 signals at the tight junction of CoPP-treated ARPE-19 cells under hyperglycemia demonstrated that the treatment could attenuate hyperglycemia-induced ROS damage at tight junctions through HO-1 induction.

The expressions of HO-1 had previously been suggested to correlate with mitochondrial respiratory activity [26]. Indeed, ARPE-19 cells treated with CoPP in either NG or HG glucose media showed significantly higher basal respiration levels than those not treated with CoPP. Likewise, the hyperglycemic ARPE-19 cells treated with CoPP displayed the highest OCR measurement compared to other groups throughout the 75 min measurement. This indicated that the CoPP-induced HO-1 expressions could promote the mitochondrial respiratory activity in ARPE-19 cells by attenuating the hyperglycemia-induced ROS damaging effect. Although increased mitochondrial respiration also correlates with an increase in viable cell numbers, further experiments are needed to demonstrate the effect was due to HO-1 activity and not a possible off-targeting effect of CoPP.

Nevertheless, it was demonstrated that the increased mitochondrial respiration from CoPP treatment was likely due to increased mitochondrial biogenesis, which is well known to be closely associated with mitochondrial respiration [27]. PGC-1α is a critical transcriptional coactivator that plays a central role in regulating mitochondrial biogenesis, as its activation leads to elevated expressions of Nrf1 and mtTFA [28]. Physiologically, the expression level of PGC-1α is low unless the cells are stimulated with external stimuli such as hyperglycemia-induced production of ROS [29]. However, some in vivo data have indicated that long-term hyperglycemia exposure would lead to downregulating mitochondrial biogenesis [30]. Likewise, the present study’s findings on the protein expressions related to mitochondrial biogenesis in ARPE-19 cells were also controversial. Although there was no difference in the PGC-1α expressions, the Nrf1 expressions in ARPE-19 cells under hyperglycemia were significantly higher than those under normoglycemia. Conversely, the expressions of mtTFA were higher in ARPE-19 cells cultured under normoglycemia than those cultured under hyperglycemia. It was suspected that the elevated expressions of NRF1 in hyperglycemic ARPE-19 cells were transient and that the long-term hyperglycemic effect would downregulate the expression of the transcription factor, as noticed in other cell types [31]. The strong expressions of mtTFA seen in normoglycemic ARPE-19 cells were likely from those stored in the mitochondria [32].

Like the effect on mitochondrial respiration, HO-1 has also been demonstrated to enhance mitochondrial biogenesis [33]. Hence, it was anticipated that CoPP could promote mitochondrial biogenesis in APRE-19 cells under hyperglycemia. As expected, CoPP treatment significantly increased the expressions of the three mitochondrial biogenetic markers in ARPE-19 cells under hyperglycemia. However, under normoglycemia, CoPP treatment significantly upregulated the expressions of PGC-1α and NRF1 but downregulated the expressions of mtTFA—this thus justified CoPP treatment elevating mitochondrial respiration in ARPE-19 cells under hyperglycemia but not normoglycemia. CoPP increased the expressions of PGC-1α and NRF1 but not mtTFA, further indicating that the bioactivity of this chemical molecule could be affected by glucose concentration.

## 5. Conclusions

The present study indicated that CoPP could attenuate hyperglycemia-induced cell death in AREP-19 cells. Additionally, CoPP could downregulate hyperglycemia-induced expressions of pro-inflammatory and proangiogenic proteins in the cells and upregulate their mitochondrial respiration under hyperglycemia (Figure 7). Furthermore, the anti-inflammatory effects of CoPP were found to promote mitochondrial biogenesis and subsequent mitochondrial respiration. The therapeutic effects of CoPP were likely mediated through the induced HO-1 expressions, although other possible off-targeting effects cannot be ruled out. Further studies are required to understand the impact of different glucose concentrations on the bioactivity of CoPP, as well as the long-term effect of exposing the cells to the drugs. Moreover, it would be more clinically relevant if the study could be repeated with primary or induced pluripotent stem cell-derived RPE cells, rather than human cell lines, which is a limitation of the present study.

## Figures and Tables

**Figure 1 antioxidants-14-00092-f001:**
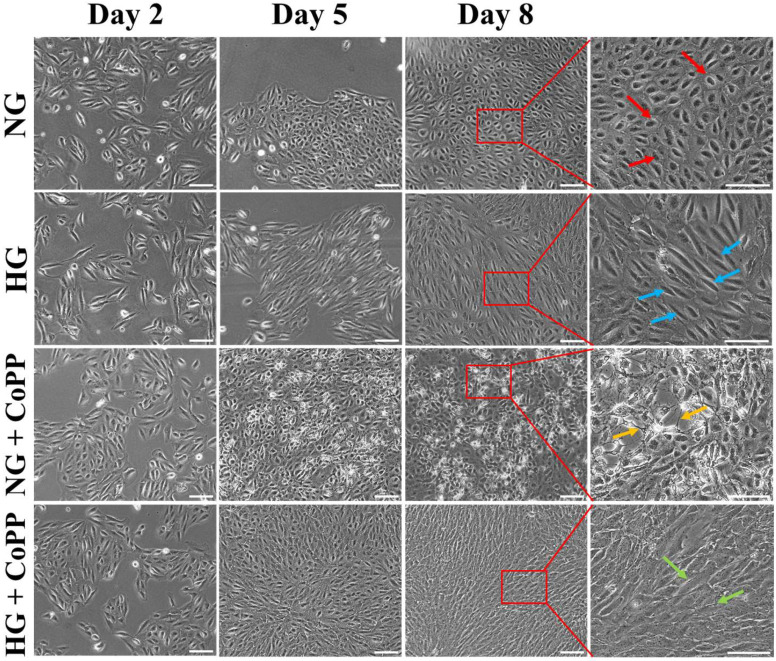
Images of ARPE-19 cells cultured under normoglycemia or hyperglycemia, with or without CoPP. ARPE-19 cells were cultured in media containing either 5 mM glucose (NG) or 25 mM glucose (HG). The results were compared to those that had 0.1 µM of CoPP. The red, blue, yellow, and green arrows indicated morphological differences that were noticed in the cell cultures. The areas of the cell culture in red boxes were enlarged. Scale bar, 50 μm.

**Figure 2 antioxidants-14-00092-f002:**
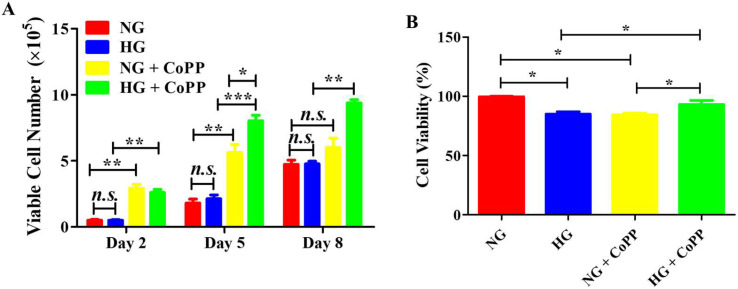
Total viable cell counts and cell viability. ARPE-19 cells were cultured continuously for 8 days in NG or HG media, with or without CoPP. Media were changed every three days, (**A**) and the numbers of viable cells were counted. (**B**) The cell viability on day 8 was determined for each group. *, *p* < 0.05; **, *p* < 0.01; ***, *p* < 0.001; *n.s.*, not significant.

**Figure 3 antioxidants-14-00092-f003:**
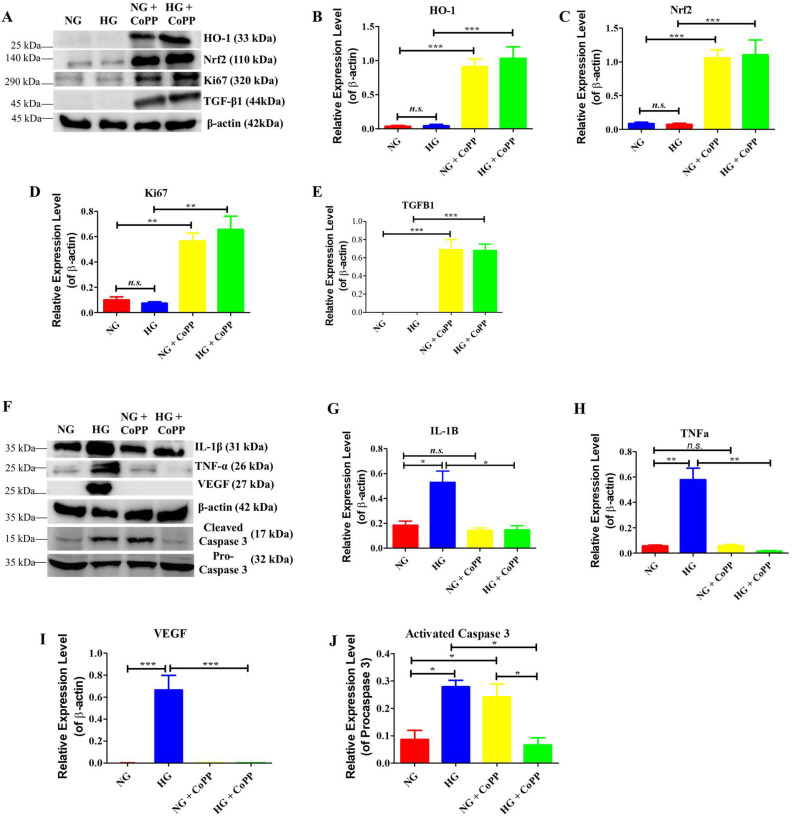
Protein expressions in CoPP-treated ARPE-19 cells. Cell lysates were extracted from ARPE-19 cells cultured for 8 days under normoglycemia or hyperglycemia, with or without CoPP treatment. The protein samples were prepared and subjected to Western blotting analysis for (**A**) the expressions of HO-1/Nrf2 proteins and proliferation protein markers. The expression levels were quantified with the Odyssey Fc imaging system (**B**–**E**). Likewise, the expression levels of inflammatory and apoptotic protein markers in the extracted samples were also analyzed and quantified (**F**–**J**). N = 3, *n.s.*, not significant; *, *p* < 0.05; **, *p* < 0.01; ***, *p* < 0.001.

**Figure 4 antioxidants-14-00092-f004:**
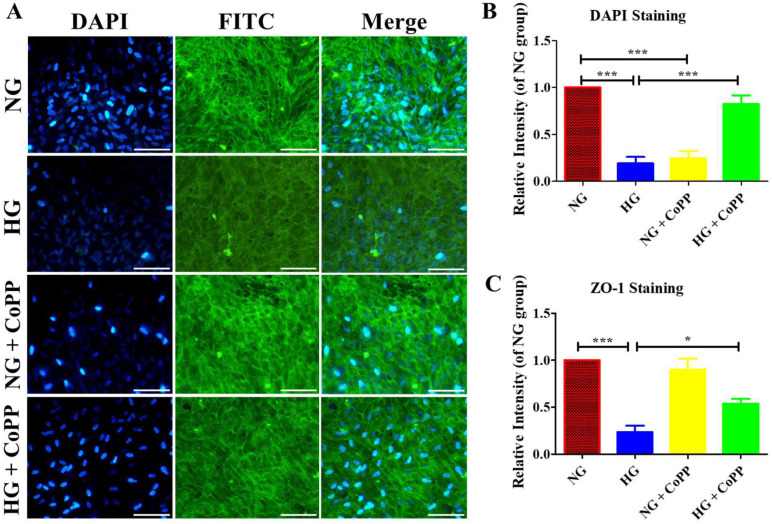
Immunostaining of ZO-1 expressions in CoPP-treated cells. (**A**) The ARPE-19 cells were cultured in either NG or HG, with or without CoPP, for 8 days before subjecting to DAPI (blue) and ZO-1 staining (green). Scale bar, 50 µm. The measured fluorescent intensities (**B**) DAPI and (**C**) ZO-1 staining in each treatment group were normalized against the NG group. N = 3; *, *p* < 0.05; ***, *p* < 0.001.

**Figure 5 antioxidants-14-00092-f005:**
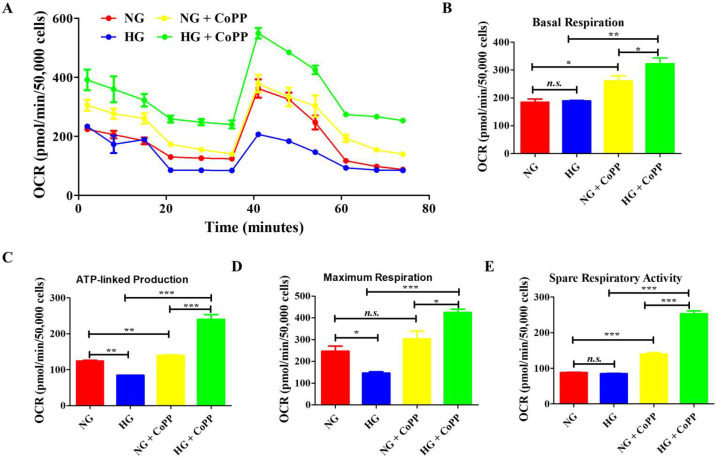
The effects of CoPP on mitochondrial respirations in ARPE-19 cells. (**A**) The kinetic profile of ARPE-19 cells. Based on the kinetic profile, the following were calculated: (**B**) basal respiration, (**C**) ATP-linked production, (**D**) maximum respiration, and (**E**) spare respiratory activity. N = 5, *, *p* < 0.05; **, *p* < 0.01; ***, *p* < 0.001; *n.s.*, not significant.

**Figure 6 antioxidants-14-00092-f006:**
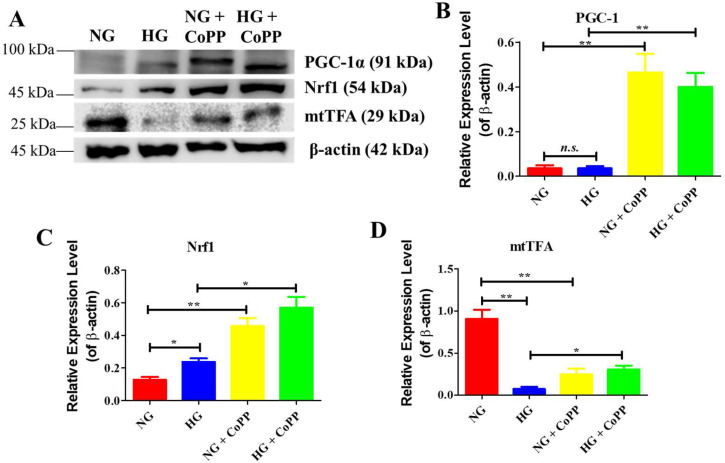
The effects of CoPP on mitochondrial biogenesis. (**A**) Proteins related to mitochondrial biogenesis were analyzed by Western blotting, and (**B**–**D**) the relative expression levels were quantified and statistically analyzed. N = 3, *n.s*., not significant; *, *p* < 0.05; **, *p* < 0.01.

**Figure 7 antioxidants-14-00092-f007:**
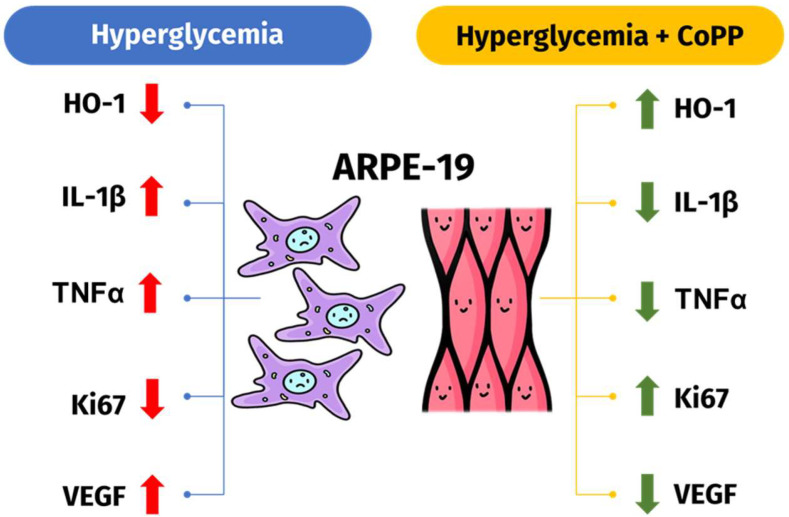
CoPP downregulates the expressions of pro-inflammatory and pro-angiogenic markers in ARPE-19 cells under hyperglycemia. CoPP treatments downregulated hyperglycemia-induced inflammatory proteins’ expressions, promoting the cells’ mitochondrial respiration and biogenesis. CoPP was also demonstrated to downregulate VEGF expressions in the cells under hyperglycemia.

## Data Availability

The data supporting this study’s findings are available from the corresponding author upon reasonable request.

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
