# Peer review of "Cobalt Protoporphyrin Downregulates Hyperglycemia-Induced Inflammation and Enhances Mitochondrial Respiration in Retinal Pigment Epithelial Cells"

_antioxidants, 2025, doi:10.3390/antiox14010092_

Round 1

Reviewer 1 Report

In the manuscript entitled ¨Cobalt Protoporphyrin Downregulates Hyperglycemia-Induced 2 Inflammation and Enhances Mitochondrial Respiration in Retinal Pigment Epithelial Cells¨ authors analyze the effects of CoPP in retinal cells under normoglycemia or hyperglycemia, concluding that the CoPP addition in hyperglycemic medium improves the mitochondrial biogenesis, respiration with a protective effect. Nevertheless, major revision is needed to clarify the results.

  • In the abstract, describe the meaning of CoPP.
  • The introduction needs improvement; now, only a brief paragraph corresponds to CoPP. Please clarify the relevance of mitochondrial respiration and mitochondrial biogenesis in RPE cells in the context of diabetic retinopathy.
  • Rename section 2.3 only a WB of
  • In lines 85 and 99, change the @m to µm.
  • In line 100 defines the meaning of TBST.
  • Did the authors determine the glucose concentrations in the medium? Or indicate the medium changes and glucose alimentation made in the 8 days? This is relevant to demonstrate the maintenance of the cell hyperglycemia in the experiment days. This clarification is necessary.
  • Did the authors determine the lactate production in the NG and HG cultures? This clarification is necessary to assess mitochondrial respiration.
  • In section 2.4, indicates the function of mitochondrial respiration of oligomycin, FCCP, antimycin, and rotenone. This clarification is desirable.
  • In line 108, correct reference 14 following the journal indications.
  • Please include the incise in each image in the supplementary file (A, B, C, etc.) (Figure S1C-D). This clarification is necessary.
  • Following the journal format, please move the images after the corresponding result description. This modification is desirable.
  • In line 166, the authors declare ¨Surprisingly, CoPP treatment significantly attenuated the expressions of activated Caspase-3 in ARPE-19 cells under hyperglycemia but displayed high activated caspase-3 expressions in the cells under normoglycemia (Figure S1H).¨Nevertheless, the image in S1  no corresponds to authors description. In Figure S1, only optical images are shown; the reader cannot determine the presence (or absence) of caspase-3 expression. This clarification is necessary.
  • In Figure 5, the color assigned to NG and NG+CoPP groups must be changed. Graphics 5A and 5B are confusing, and the same is true in Figure 6B. This clarification is necessary.
  • Should the entire western blot images be included in the ¨original image file before being cropped?
  • In the conclusion section, please include the limitations of the manuscript. 

Author Response

Major Comments
In the manuscript entitled ¨Cobalt Protoporphyrin Downregulates Hyperglycemia-Induced Inflammation and Enhances Mitochondrial Respiration in Retinal Pigment Epithelial Cells¨ authors analyze the effects of CoPP in retinal cells under normoglycemia or hyperglycemia, concluding that the CoPP addition in hyperglycemic medium improves the mitochondrial biogenesis, respiration with a protective effect. Nevertheless, major revision is needed to clarify the results.

Detail Comments

Comment 1: In the abstract, describe the meaning of CoPP.

Response 1: We thank the reviewer for pointing out the mistake. The following was added in line 13: “Cobalt protoporphyrin”

Comment 2: The introduction needs improvement; now, only a brief paragraph corresponds to CoPP. Please clarify the relevance of mitochondrial respiration and mitochondrial biogenesis in RPE cells in the context of diabetic retinopathy.

Response 2: We thank the reviewer for the comment. From Lines 64 to 70, we have outlined the significance of CoPP in relation to anti-inflammation and the biological evidence provided by others. We did not go into detail on the chemical structure of CoPP, as we believe it is not the focus of this study. Moreover, there are already many review papers in the literature that cover the chemical structure of CoPP.
As for the relevance of mitochondrial respiration and biogenesis in RPE cells in the context of DR, we have added the following sentence in Line 36: “Additionally, mitochondrial respiration and biogenesis are downregulated in DR patients' RPE cells”

Comment 3: Rename section 2.3 only a WB of

Response 3: We thank the reviewer for pointing out the mistake. We have reworded the title of this section to “Dodecyl Sulfate Polyacrylamide Electrophoresis (SDS-PAGE) and Western blotting”

Comment 4: In lines 85 and 99, change the @m to µm.

Response 4: We thank the reviewer for pointing out the mistake. We have corrected the mistake and highlighted the corrections in yellow.

Comment 5: In line 100 defines the meaning of TBST.

Response 5: We thank the reviewer for pointing out the mistake. We have added the following words before the acronym: “0.1% Tris-buffered saline with 0.1% Tween® 20 detergent (TBST).”

Comment 6: Did the authors determine the glucose concentrations in the medium? Or indicate the medium changes and glucose alimentation made in the 8 days? This is relevant to demonstrate the maintenance of the cell hyperglycemia in the experiment days. This clarification is necessary.

Response 6: We thank the reviewer for the comment. The experiment used commercially available media containing either 5.5 mM glucose or 25 mM glucose. Media in every culture sample were changed every 2-3 days. which was indicated in Section 2.1, line 79.

Comment 7: Did the authors determine the lactate production in the NG and HG cultures? This clarification is necessary to assess mitochondrial respiration.

Response 7: We thank the reviewer for the comment. We did not check for lactate production, as a Seahorse metabolic analyzer was used to determine the changes in mitochondrial respiration.

Comment 8: In section 2.4, indicates the function of mitochondrial respiration of oligomycin, FCCP, antimycin, and rotenone. This clarification is desirable.

Response 8: We thank the reviewer for the comment. We have added the following sentences to Section 2.4:
Oligomycin inhibits ATP synthase (complex V) that reduces mitochondrial respiration linked to cellular ATP production. FCCP is an uncoupling agent that collapses the proton gradient and disrupts the mitochondrial membrane potential. It was injected after oligomycin. As a result, electron flow through the ETC was uninhibited, and oxygen consumption by complex IV reached the maximum. The FCCP-stimulated oxygen consumption rate could then be used to calculate spare respiratory capacity. The last injection was a mixture of rotenone, a complex I inhibitor, and antimycin A, a complex III inhibitor. This combination shuts down mitochondrial respiration and enables the calculation of nonmitochondrial respiration driven by processes outside the mitochondria.

Comment 9: In line 108, correct reference 14 following the journal indications.

Response 9: We thank the reviewer for pointing out the mistake. The correction was made and highlighted in yellow.

Comment 10: Please include the incise in each image in the supplementary file (A, B, C, etc.) (Figure S1C-D). This clarification is necessary.

Response 10: We apologize for the poor labeling mistake and have corrected it. The statistical analysis of each western blot data set is placed next to the images in the main text rather than moved to the supplementary.

Comment 11: Following the journal format, please move the images after the corresponding result description. This modification is desirable.
Response 11: We thank the reviewer for the recommendation. We have re-formatted the manuscript and moved each figure behind their respective section.

Comment 12: In line 166, the authors declare ¨Surprisingly, CoPP treatment significantly attenuated the expressions of activated Caspase-3 in ARPE-19 cells under hyperglycemia but displayed high activated caspase-3 expressions in the cells under normoglycemia (Figure S1H).¨Nevertheless, the image in S1  no corresponds to authors description. In Figure S1, only optical images are shown; the reader cannot determine the presence (or absence) of caspase-3 expression. This clarification is necessary.

Response 12: We apologize for the confusion. As mentioned in the response to Comment 10, we made a huge mistake with the labeling in Figure 3 and the corresponding citation in the main text. The mistake has been corrected. The statistical analysis for the western blotting of activated caspase-3 is Figure 3J, which shows the addition of CoPP downregulated the expressions in hyperglycemic ARPE-19s. In contrast, CoPP elevated the activated caspase-3 expressions in normoglycemic ARPE-19 cells.

Comment 13: In Figure 5, the color assigned to NG and NG+CoPP groups must be changed. Graphics 5A and 5B are confusing, and the same is true in Figure 6B. This clarification is necessary.

Response 13: We apologize for the confusion and have changed all the colors to the bar graph.

Comment 14: Should the entire western blot images be included in the ¨original image file before being cropped?

Response 14: We thank the reviewer for the comment. Due to the large number of markers being studied, the membranes were cut according to the expected size. The data presented in the original image file were the images before being cropped.

Comment 15: In the conclusion section, please include the limitations of the manuscript.

Response 15: We thank the reviewer for the suggestion. We have added the following sentence in the Conclusion: “Moreover, it would be more clinically relevant if the study could be repeated with primary or induced pluripotent stem cells-derived RPE cells rather than human cell lines, which is a limitation of the present study.”

Reviewer 2 Report

The manuscript is highly relevant for the field of DR. The text is well written. 

The manuscript must be organized in a must better format because Figures should appear after or within each section. 

Images are all together and it makes difficult the results intrepretation. Images should be close each section in the text. 

line 85 "10...M" I am not familiar with the symbol after the number 10. 

Line 88 - "4C" shoul be 4ºC

Line 122 - ND and HG should be defined at this point

Line 387 - First, in the conclusion section Figures should not be referenced . Then, this Figure 6 should be Figure 7. 

Author Response

Major Comments

Comment 1: The manuscript is highly relevant for the field of DR. The text is well written.

Response 1: We thank the reviewer for the kind response.

Comment 2: The manuscript must be organized in a must better format because Figures should appear after or within each section

Response 2: We thank the reviewer for the recommendation. We have re-formatted the manuscript and moved each figure behind their respective section.

Detail Comments

Comment 1: Images are all together and it makes difficult the results intrepretation. Images should be close each section in the text.

Response 1: We apologize for the inconvenience. We have re-formatted the manuscript and moved each figure behind their respective section.

Comment 2: line 85 "10...M" I am not familiar with the symbol after the number 10.

Response 2: We apologize for the mistake. The symbol has been corrected. It should be “0.1 µM”. We have highlighted the correction in yellow.

Comment 3: Line 88 - "4C" shoul be 4ºC

Response 3: We thank the reviewer for pointing out the mistake. We have corrected the mistake and highlighted the correction in yellow.

Comment 4: Line 122 - ND and HG should be defined at this point

Response 4: We thank the reviewer for pointing out the mistake. We have corrected the sentences to: “5.5 mM (normoglycemia, NG) or 25 mM (hyperglycemia, HG) glucose”. The correction is highlighted in yellow.

Comment 5: Line 387 - First, in the conclusion section Figures should not be referenced . Then, this Figure 6 should be Figure 7. 

Response 5: We thank the reviewer for the comment and for pointing out our typing error. We agree it is unusual to include a figure in the conclusion section. However, given the biological complexity that glucose had on ARPE-19 cells and the function of CoPP, we felt it would help readers understand the “take-home” message by referencing a schematic diagram in the conclusion section. We have changed the numbering to “Figure 7”. We have highlighted the changes in yellow.

Reviewer 3 Report

The present experimental study is interesting. The authors investigated the efficacy of cobalt protoporphyrin in Retinal Pigment Epithelial Cells.  Results of animal studies indicate that heme oxygenase (HO) expression and activity are downregulated in experimentally induced diabetes. That is associated with severe hormonal and metabolic disturbances. However, these pathological changes are reversed by therapy with HO activators. In animals with experimentally induced diabetes, HO was upregulated by genetic manipulation or pharmacological activators such as hemin and cobalt protoporphyrin.

The authors' choice of the dose of 0.1 μM of cobalt protoporphyrin and the hyperglycemic state of 25 mM (HG) glucose is a crucial aspect of their study design. This choice is likely based on their understanding of the compound's therapeutic potential and its association with human therapeutic regimens. However, the authors could provide more explicit justification for these specific conditions.

The authors have provided a comprehensive and detailed description of the pathways involved in their study. This thoroughness in explaining the methodology enhances the reader's understanding of the research process and the mechanisms at play. The manuscript is well-written, and the discussion and conclusions are acceptable.

Overall, the data are of interest and underscore the value of further research in this area.

none

Author Response

Major comments

The present experimental study is interesting. The authors investigated the efficacy of cobalt protoporphyrin in Retinal Pigment Epithelial Cells. Results of animal studies indicate that heme oxygenase (HO) expression and activity are downregulated in experimentally induced diabetes. That is associated with severe hormonal and metabolic disturbances. However, these pathological changes are reversed by therapy with HO activators. In animals with experimentally induced diabetes, HO was upregulated by genetic manipulation or pharmacological activators such as hemin and cobalt protoporphyrin.

Comment 1: The authors' choice of the dose of 0.1 μM of cobalt protoporphyrin and the hyperglycemic state of 25 mM (HG) glucose is a crucial aspect of their study design. This choice is likely based on their understanding of the compound's therapeutic potential and its association with human therapeutic regimens. However, the authors could provide more explicit justification for these specific conditions.
Response 1: We thank the reviewer for the comment. The concentration was chosen based on the results from one of our previous studies. We have added the following sentence in Section 2.1: “The working concentration of 0.1 µM was identified as the best concentration to work with, according to a previous study on the therapeutic effect of CoPP in keratinocytes”.

Comment 2: The authors have provided a comprehensive and detailed description of the pathways involved in their study. This thoroughness in explaining the methodology enhances the reader's understanding of the research process and the mechanisms at play. The manuscript is well-written, and the discussion and conclusions are acceptable.

Response 2: We thank the reviewer for the positive comment.

Comment 3: Overall, the data are of interest and underscore the value of further research in this area.

Response 3: We thank the reviewer for the positive comment.

Round 2

Reviewer 1 Report

The authors respond to all the reviewers' suggestions, such as improving the graph color and adding a better description of the methods. A better description of the methodological data allows readers to replicate the experiments and advance their discoveries. The conclusion section will also include the study's limitations.

After revising the new manuscript version, the reviewer confirmed that the manuscript was ready to publish.

After revising the new manuscript version, the reviewer confirmed that the manuscript was ready to publish.